# Understanding opportunities for efficiency in single-image super resolution networks

## Abstract

A successful application of convolutional architectures is to increase the resolution of single low-resolution images – a image restoration task called super-resolution (SR). Naturally, SR is of value to resource constrained devices like mobile phones, electronic photograph frames and televisions to enhance image quality. However, SR demands perhaps the most extreme amounts of memory and compute operations of any mainstream vision task known today, preventing SR from being deployed to devices that require them. In this paper, we perform a early systematic study of system resource efficiency for SR, within the context of a variety of architectural and low-precision approaches originally developed for discriminative neural networks. We present a rich set of insights, representative SR architectures, and efficiency trade-offs; for example, the prioritization of ways to compress models to reach a specific memory and computation target and techniques to compact SR models so that they are suitable for DSPs and FPGAs. As a result of doing so, we manage to achieve better and comparable performance with previous models in the existing literature, highlighting the practicality of using existing efficiency techniques in SR tasks. Collectively, we believe these results provides the foundation for further research into the little explored area of resource efficiency for SR.

## 1 Introduction

Rapid progress has been made in the development of convolutional networks (Dong et al., 2015) that are capable of taking a low-resolution image and producing an image with a significant increase in resolution. This image restoration task is referred to as super-resolution (SR) and has many potential applications in devices with limited memory and compute capacity. The fundamental problem however is that the state-of-the-art networks (Lim et al., 2017; Zhang et al., 2018; Zhang et al., 2018) consist of thousands of layers and are some of the most resource intensive networks currently known. Furthermore, due to the spatial dimensions of feature maps needed to maintain or up-scale the input, the number of operations are counted in the billions as opposed to millions in models for discriminative tasks. As a result, there is a need for a general systematic approach to improve the efficiency of SR models.

The challenge of the system resource requirements for deep learning models for tasks other than SR have been carefully studied in previous works (Zhang et al., 2017b; Howard et al., 2017; Ma et al., 2018; Sandler et al., 2018), achieving massive gains in size and compute with little to no loss in performance. These reductions are achieved with a wide variety of methods being developed grounded in primarily architecture-level changes and techniques grounded in the use of low precision and quantized model parameters. However, how these efficiency methods behave when applied within SR have not yet been studied in significant depth, with very few results published in the literature. Extrapolating from prior results for other tasks is problematic given that predominantly existing studies are applied to discriminative tasks with substantially different architectures and operations. Due to the up-sampling structure of SR models, these efficiency methods may therefore produce potentially stronger side-effects to image distortion.

In this paper, we detail a systematic study that seeks to bridge current understanding in SR and known approaches for scaling down the consumption of system resources by deep models. By

examining the impact on image distortion quality when performing various efficiency techniques, we provide the following new insights:

- The effectiveness of low rank tensor decomposition and other convolution approximations, which are comparable and successful in discriminative tasks, can vary considerably in SR. (See section 4.1).

- Unlike image discriminative networks, SR networks suffer from a worse trade-off between efficiency and performance as more layers are compressed. (See section 4.2)

- The practicality of adopting compression techniques for other tasks to SR as our best models are better or comparable to existing literature. For instance, our best model achieves significantly better performance and 6x less compute than MemNet (Tai et al., 2017b) and VDSR (Kim et al., 2015b). Additionally, it also performs better and is 4.1x-5.8x smaller than SRMDNF (Zhang et al., 2017a). (See section 4.3)

- Successful quantization techniques used in image discriminative tasks are equally successful in SR. (See section 5)

## 2    RELATED WORK

We focus on using neural networks for SR as they have shown to achieve superior performance against previous traditional approaches (Timofte et al., 2013; Kim & Kwon, 2010; Chang et al., 2004). An SR image can either be evaluated using standard image distortion metrics, such as PSNR, SSIM (Wang et al., 2004) and IFC (Sheikh et al., 2005), or using perception metrics, such as Ma et al. (2016), NIQE (Mittal et al., 2013), and BRISQUE (Mittal et al., 2012). Blau & Michaeli (2017) provided theoretical backups on the trade-off between image distortion and perception.

**Distortion SR:** In the distortion line of work, models favour pixel-to-pixel comparisons and are usually trained on either the L1 or L2 (MSE) loss. These models have been known to produce more visually pleasing outcomes on structural images Blau et al. (2018) than perceptual SR models. Dong et al. (2015) first proposed using convolutional networks for SR, leading to a surge in using neural networks for SR. These networks differ in their building blocks for feature extraction and up-sampling. For instance, Dong et al. (2016) proposed a faster convolutional network by taking the down-sampled low-resolution image as an input. Other variations include using more layers (Kim et al., 2015b), recursive layers (Kim et al., 2015a; Tai et al., 2017a), memory blocks (Tai et al., 2017b; Ahn et al., 2018), DenseNet (Huang et al., 2016) blocks (Tong et al., 2017), residual (He et al., 2015) blocks (Ledig et al., 2016; Lim et al., 2017; Kim & Lee, 2018), and multiple-image degradations (Zhang et al., 2017a). Additionally, more recent models use attention (Bahdanau et al., 2014) mechanisms (Liu et al., 2018; Zhang et al., 2018), back-projection (Haris et al., 2018; Navarrete Michelini et al., 2018), and other non-conventional non-linear layers (Choi & Kim, 2017; Gu et al., 2018).

**Perceptual SR:** Perceptual SR models, on the other hand, are better at reconstructing unstructured details with high perceptual quality (Blau et al., 2018). These models usually adopt popular models for image distortion and train them using a variety of different loss functions, such as the perceptual loss (Johnson et al., 2016), contextual loss (Mechrez et al., 2018b), adversarial loss (Goodfellow et al., 2014), and the Gram loss (Gatys et al., 2015). For instance, Choi et al. (2018) adopted EUSR Kim & Lee (2018) and Ledig et al. (2016); Wang et al. (2018); Mechrez et al. (2018a) adopted SRResNet (Ledig et al., 2016) by making slight architecture changes and replacing the objective. Although these perceptual models are able to generate more visually pleasing results on certain images, they do not seem to work well as inputs for image classification (Jaffe et al., 2017).

**Efficient SR:** As models in both tracks are resource-intensive, the recent PIRM 2018 Challenge for mobile (Ignatov et al., 2018) presented a range of high efficiency models that were designed to run faster and perform better than SRCNN (Dong et al., 2015). These models are complementary to our work and can follow our best practices to achieve greater efficiency gains. A work closely related to our work is done by Ahn et al. (2018) who systemically investigate the impact of using grouped convolutions. Due to the massive design space caused by the variability of training and evaluating these models, we focus on the trade-offs between performance measured by the image distortion metrics and efficiency and leave the rest as future work.

## 3    SYSTEMATIC STUDY OF LOW-RESOURCE SUPER RESOLUTION NETWORKS

The key step in our work is to build understanding towards building resource-efficient architectures for super-resolution. While there is a lot of understanding of how these efficiency-saving techniques work in classification problems, there is a lack of experimental studies and systematic approaches to understand their practicality in super-resolution. To our knowledge, this is the first systematic study of wide range efficiency methods on super-resolution.

We measure performances using PSNR and SSIM (Wang et al., 2004) and measure efficiency of memory and compute using the number of parameters and the number of multiply-add operations (Mult-Adds), both of which dictate which platform these models can run on. However, these metrics alone do not reflect the trade-off between performance and efficiency. Therefore, we introduce two new metrics that measures the number of Giga Mult-Adds saved and the number of parameters saved for every 0.01dB PSNR loss in the test sets: Set5 (Bevilacqua et al., 2012), Set14 (Yang et al., 2010), B100 (Martin et al., 2001), and Urban100 (Huang et al., 2015). These metrics are calculated by taking the difference between the compressed model and the uncompressed model. All Mult-Adds are calculated by upscaling to a 720p image.

We decide to use RCAN Zhang et al. (2018) as our baseline model as it proves to be the state-of-the-art and has the best performance in the image distortion metrics at the time of writing. We take its simplest building block and build a shallower network and use that as a basis for exploring the use of a variety of techniques.

**Implementation Details:** We train our models in section 4 and section 5.1 in the same manner as that of EDSR Lim et al. (2017). In particular, we use $48 \times 48$ RGB patches of LR images from the DIV2K dataset Timofte et al. (2017). We augment the training data with random horizontal flips and 90 degree rotations and pre-process them by subtracting the mean RGB value of the DIV2K dataset. Our model is trained using the ADAM Kingma & Ba (2014) optimizer with hyper-parameters $\beta_1 = 0.9, \beta_2 = 0.999$, and $\epsilon = 10^{-8}$. The mini-batch size is 16, learning rate begins with $1e-4$ and is halved at 200 epochs, and the model is trained for 300 epochs using L1 loss. We train x2 models from scratch and use them as pre-trained models to train x3 and x4 models for faster convergence. Lastly, for ternary quantization in section 5.2, we further train the model with quantization enabled in each forward pass for 40 epochs, starting at a learning rate of $5e-5$, and then fix the quantized ternary weights and further train for another 10 epochs at a learning rate of $2.5e-5$.

## 4    EFFICIENT NETWORK ARCHITECTURES FOR SUPER RESOLUTION

We begin our evaluation by conducting a series of experiments: (i) we explore the effects of applying different resource-efficient architectures to our baseline model (section 4.1), (ii) we consider two best techniques and present trade-off solutions while applying them to different parts of our baseline model (section 4.2), (iii) and lastly, we compare our best results with previous SR architectures (section 4.3).

### 4.1    EFFECTS OF VARIOUS RESOURCE-EFFICIENT TECHNIQUES

**Motivation:** Resource-efficient architectures use various low rank tensor decomposition and other convolutional approximation techniques, which is agnostic and is not specifically designed for any particular task, to build fast and accurate image discriminative models. We first develop an initial understanding of the trade-off solution by replacing and modifying 3x3 convolution layer blocks in the baseline model.

**Approach:** We explore the use of known techniques such as the bottleneck design, separable/grouped convolutions, and channel shuffling. We take the feature extraction unit from resource-efficient architectures and remove all batch normalisation layers as they were previously shown to reduce performance and increase GPU memory usage (Lim et al., 2017). For our first set of experiments, we replace all 3x3 convolution layers in the residual groups of our baseline model.

*bl:* Our baseline model from RCAN (Zhang et al., 2018). We reduce the number of residual groups (RG) from 10 to 2 and the number of residual channel attention block (RCAB) in each RG from 20 to 5. We use a feature map size of 64. Making the network shallower and small in parameters allow us to clearly understand each architectural changes as opposed to having a deep network which may cause other effects and interplay.

*blrn(r):* We adopt the residual bottleneck design from ResNet (He et al., 2015) with a reduction factor of *r*. Specifically, a 1x1 convolution is used to compress information among channels by a reduction factor, resulting in a cheaper 3x3 convolution. Another 1x1 convolution is then used to recover the dimension of the output channel and a skip connection is used to pass on information that may have been lost.

*blrxn(r,g):* We replace the 3x3 convolution in *blrn* to a 3x3 grouped convolution, forming a block that is similar to that of ResNeXt (Xie et al., 2016) with an additional group size of *g*. Computation cost is further reduced by the use of grouped convolutions (Krizhevsky et al., 2012).

*blm1:* In order to further improve efficiency of the 3x3 grouped convolution, we can maximise the group size, forming a convolution that is known as depthwise convolution. Following this idea, we adopt the MobileNet v1 (Howard et al., 2017) unit which uses depthwise separable convolutions, each consist of a 3x3 depthwise convolution followed by a 1x1 convolution, also known as a pointwise convolution.

*bleff(r):* We can further approximate the 3x3 depthwise convolution by using a 1x3 and a 3x1 depthwise convolution, a technique that is used in EffNet (Freeman et al., 2018). We adopt the unit from EffNet by removing the pooling layers.

*bls1(r, g):* We group both 3x3 and 1x1 convolutions and added channel shuffling in order to improve the information flow among channels. In order to test the effects of channel shuffling, we adopt the ShuffleNet v1 (Zhang et al., 2017b) unit.

*blclc(g1, g2):* Channel shuffle is also used in Clcnet (Zhang, 2017) to further improve efficiency of *blm1*. In order to maximise efficiency from our adoption of ClcNet units, we follow the group size guidelines recommended by the authors for both the group sizes of the 3x3 (*g1*) and 1x1 (*g2*) grouped convolution.

*bls2:* Apart from using grouped convolutions, Ma et al. (2018) proposed splitting the flow into two, which is termed as channel splitting, and performing convolution on only half of the input channels in each unit at each pass. Channel shuffle is then used to enable information flow between both branches.

*blm2(e):* Inverted residuals can be used to enable skip connections directly on the bottleneck layers. Therefore, we adopt the MobileNet v2 (Sandler et al., 2018) unit in our experiments

Table 1: Quantitative results of applying resource-efficient techniques. **bold**/*italics* indicates best/second-best trade-off.

| Scale | Model | Params (K) | Mult-Adds (G) | Set5 PSNR/SSIM | Set14 PSNR/SSIM | B100 PSNR/SSIM | Urban100 PSNR/SSIM | Mult-Add Saved (G) | Params Saved (K) |
|---|---|---|---|---|---|---|---|---|---|
| | bl | 1006 | 231.2 | 37.86/0.9600 | 33.39/0.9159 | 32.06/0.8982 | 31.74/0.9248 | - | - |
| | blrn(r=2) | 464 | 106.5 | 37.61/0.9591 | 33.18/0.9137 | 31.90/0.8961 | 31.09/0.9173 | **0.9819** | **4.27** |
| | blm1 | 265 | 60.7 | 37.45/0.9586 | 33.00/0.9122 | 31.79/0.8948 | 30.65/0.9128 | *0.7894* | *3.43* |
| | blrxn(r=2, g=4) | 305 | 69.9 | 37.45/0.9585 | 33.01/0.9123 | 31.80/0.8948 | 30.71/0.9131 | 0.7755 | 3.37 |
| | blrn(r=4) | 258 | 59 | 37.42/0.9583 | 32.96/0.9120 | 31.76/0.8943 | 30.66/0.9123 | 0.7653 | 3.32 |
| 2× | blclc(g1=32, g2=2) | 232 | 52.9 | 37.40/0.9581 | 32.94/0.9118 | 31.73/0.8939 | 30.53/0.9108 | 0.7278 | 3.16 |
| | bls2 | 211 | 48.4 | 37.37/0.9580 | 32.96/0.9117 | 31.71/0.8936 | 30.49/0.9102 | 0.7254 | 3.15 |
| | bleff(r=2) | 257 | 58.7 | 37.33/0.9580 | 32.94/0.9114 | 31.71/0.8936 | 30.41/0.9098 | 0.6485 | 2.82 |
| | blm2(e=2) | 561 | 128.9 | 37.43/0.9586 | 33.08/0.9130 | 31.81/0.8949 | 30.77/0.9138 | 0.5219 | 2.27 |
| | bls1(r=2, g=4) | 188 | 42.9 | 37.06/0.9568 | 32.74/0.9096 | 31.52/0.8912 | 29.94/0.9031 | 0.4968 | 2.16 |
| | blm2(e=3) | 763 | 175.4 | 37.56/0.9589 | 33.11/0.9133 | 31.86/0.8954 | 30.93/0.9155 | 0.3509 | 1.53 |

**Results:** Our results in Table 1 show that techniques that result in a better trade-off between memory and performance will have a better trade-off between compute and performance. [1]. Overall, the use of bottlenecks alone (*blrn*) result in the best trade-offs followed by the use of separable/grouped convolutions.

Reducing the number of features to accommodate inverted bottlenecks (*blm2*) severely impact the performance and thus we omit the results from the table. We speculate that doing so would result in insufficient features at the up-sampling layer to fully capture the up-sampled image representation. Thus, we use the same number of feature maps as our bottleneck. Although the use of inverted

---

[1]Results for 3x and 4x upscaling show similar performance and efficiency trade-offs

residuals in our experiments seem worse off, it may perform better on models that use a larger feature size or multiple smaller up-sampling layers.

Lastly, the use of 1x1 grouped convolution or channel splitting with channel shuffling further reduces the evaluation metric. Although doing so can drastically reduce size, the trade-off does not seem to justify its advantages. Therefore, we recommend using bottlenecks for building resource-efficient SR architectures. If the budget for memory and efficiency is tight, we recommend the use of depthwise separable convolutions instead.

In image discriminative tasks, the proposed architecture changes are comparable in terms of efficiency and accuracy trade-offs. In our work, we show that the sole use of low rank tensor decomposition (bottleneck architectures) provide the best trade-offs, followed by the use of separable/grouped convolutions and the use of both channel splitting and shuffling.

## 4.2    EFFECTS OF ARCHITECTURAL LAYERS BETWEEN THE INPUT AND OUTPUT LAYER

**Motivation:**   Bhattacharya & Lane (2016) and Kim et al. (2015c) have shown that it is possible in image classification to maintain a similar or slight drop in performance by decomposing tensors of known models. However, our models suffer a significant drop in performance. (Table 1). Therefore, in order to further understand the extent of their applicability in SR, we apply the top two best techniques, which are bottleneck reduction (*blrn*) and depthwise separable convolutions (*blm1*), on various different parts of our baseline model.

**Approach:**   Our preliminary experiments with applying some of these techniques on the first and last convolution layer led to worse trade-offs. Therefore, we apply our techniques between them. We replace the sub-pixel convolution upsampling layer to the enhanced upscaling module (EUM) as proposed by Kim & Lee (2018) to allow the use of skip connections. Using EUM leads to an increase in performance at a slight cost of both memory and compute. Thus, in order to maintain the memory cost, we use recursion, forming the enhanced recursive upscaling module (ERUM) shown in figure 1. The number of ERUMs is the same as the scaling factor and each ERUM recurses twice or thrice for x2, x4 or x3 scales respectively. Experiments that use ERUMs for up-sampling are indicated with a postfix *-e*. We calculate our trade-off metrics based on our baseline model with ERUM as its up-sampling layer instead *bl-e*. We modify all 3x3 convolution layers as such:

*rb:* Changes are made in residual blocks/modules.

*rg:* Changes are made in residual groups, therefore including those in *rb*. (Experiments in section 4.1 are done in this setting.)

*rg+u:* Changes are made in both *rg* and the up-sampling layers (ERUMs).

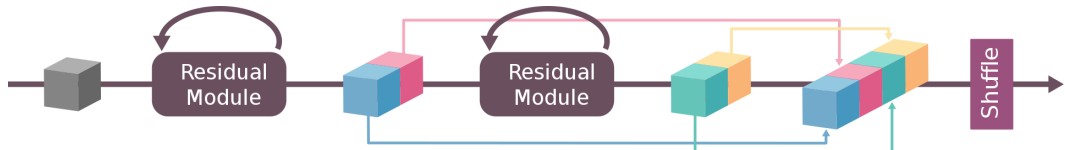

Figure 1: Our proposed ERUM for image upscaling.

**Results:**   Our results in Table 2 reinforce our findings in section 4.1 that the adoption of bottleneck reduction alone leads to the best trade-offs, followed by the use of group convolutions. Therefore, we recommend taking gradual steps to compress the model. For instance, we suggest gradually changing convolutions to use bottleneck reduction, avoiding the up-sampling, first, and last convolutions until a budget is reached. If further compression is needed, we suggest changes to the up-sampling layer or the use of group convolutions.

## 4.3    COMPARISONS WITH PREVIOUS SR MODELS

We take our derived best models based on different budgets from our first two experiments (See section 4.1 & 4.2) and compare them with the existing literature, which is shown in Table 3. For fair comparisons, we omit models that are way bigger by several magnitudes as their performances are much better. Likewise, we exclude models that are way smaller as their performance are much

Table 2: Quantitative results of applying techniques on different parts of the model. We took the best three derived models given three different budgets and compared them with previous models in Table 3 **bold**/*italics* indicates best/second-best trade-off.

| Scale | Model [Changes] | Params (K) | Mult-Adds (G) | Set5 PSNR/SSIM | Set14 PSNR/SSIM | B100 PSNR/SSIM | Urban100 PSNR/SSIM | Mult-Add Saved (G) | Params Saved (K) |
|---|---|---|---|---|---|---|---|---|---|
| | bl-e | 1006 | 265.3 | 37.92/0.9602 | 33.43/0.9162 | 32.09/0.8988 | 31.83/0.9256 | - | - |
| | blrn-e(r=2)[rb] | 535 | 156.8 | 37.75/0.9596 | 33.30/0.9153 | 32.00/0.8973 | 31.48/0.9218 | **1.4662** | **6.3649** |
| | blm1-e[rb] | 363 | 117 | 37.65/0.9592 | 33.19/0.9143 | 31.92/0.8964 | 31.13/0.9181 | *1.0746* | *4.6594* |
| 2× | blm1-e[rg] | 265 | 94.7 | 37.59/0.9590 | 33.12/0.9132 | 31.87/0.8959 | 30.91/0.9158 | 0.9584 | 4.1629 |
| | blrn-e(r=2)[rg] | 464 | 140.5 | 37.64/0.9592 | 33.20/0.9142 | 31.93/0.8967 | 31.18/0.9185 | 0.9455 | 4.1061 |
| | blrn-e(r=2)[rg+u] | 370 | 97.1 | 37.56/0.9589 | 33.10/0.9131 | 31.86/0.8954 | 30.99/0.9164 | 0.9557 | 3.6136 |
| | blm1-e[rg+u] | 137 | 35.4 | 37.35/0.9580 | 32.94/0.9117 | 31.72/0.8939 | 30.45/0.9103 | 0.8181 | 3.0925 |
| | bl-e | 1080 | 156.3 | 34.35/0.9269 | 30.29/0.8415 | 29.06/0.8046 | 28.13/0.8521 | - | - |
| | blrn-e(r=2)[rb] | 609 | 108.1 | 34.19/0.9257 | 30.18/0.8394 | 29.00/0.8026 | 27.89/0.8469 | **0.8456** | **8.2632** |
| | blm1-e[rb] | 437 | 90.5 | 34.09/0.9249 | 30.13/0.8379 | 28.95/0.8015 | 27.69/0.8419 | 0.6784 | *6.6289* |
| 3× | blrn-e(r=2)[rg+u] | 397 | 57.6 | 33.97/0.9236 | 30.04/0.8362 | 28.89/0.7997 | 27.50/0.8376 | *0.6902* | 4.7762 |
| | blrn-e(r=2)[rg] | 538 | 100.9 | 34.13/0.9251 | 30.14/0.8380 | 28.97/0.8016 | 27.72/0.8421 | 0.6368 | 6.2299 |
| | blm1-e[rg] | 339 | 80.6 | 34.08/0.9244 | 30.03/0.8356 | 28.92/0.8005 | 27.55/0.8386 | 0.6056 | 5.928 |
| | blm1-e[rg+u] | 146 | 21.4 | 33.73/0.9221 | 29.83/0.8320 | 28.78/0.7970 | 27.10/0.8283 | 0.5644 | 3.9079 |
| | bl-e | 1154 | 135.5 | 32.08/0.8942 | 28.58/0.7815 | 27.56/0.7360 | 26.16/0.7872 | - | - |
| | blrn-e(r=2)[rb] | 683 | 108.3 | 32.10/0.8938 | 28.51/0.7795 | 27.51/0.7340 | 25.95/0.7808 | **0.8774** | **15.1935** |
| | blm1-e[rb] | 511 | 98.4 | 31.98/0.8921 | 28.45/0.7778 | 27.47/0.7328 | 25.80/0.7754 | 0.5456 | *9.4559* |
| 4× | blrn-e(r=2)[rg+u] | 424 | 50 | 31.84/0.8898 | 28.34/0.7750 | 27.38/0.7295 | 25.52/0.7673 | *0.6577* | 5.6154 |
| | blm1-e[rg] | 413 | 92.8 | 32.02/0.8921 | 28.44/0.7765 | 27.44/0.7310 | 25.67/0.7715 | 0.5272 | 9.1481 |
| | blrn-e(r=2)[rg] | 612 | 104.3 | 32.02/0.8925 | 28.47/0.7779 | 27.48/0.7323 | 25.79/0.7755 | 0.5032 | 8.7419 |
| | blm1-e[rg+u] | 156 | 18.7 | 31.47/0.8847 | 28.12/0.7697 | 27.26/0.7252 | 25.16/0.7538 | 0.4928 | 4.211 |

worse. Regardless, our techniques can be applied to any model for further trade-offs between performance and efficiency.

Although our main objective is not to beat previous models but to understand and recommend techniques that can be applied to any existing model, we manage to derive models that are better or comparable to other models in the literature. For instance, in terms of size and evaluation metric, our best model (*blrn-e[rb]*) outperforms all models that have a count of 1,500K parameters and below. By comparing compute and evaluation, our best model performs better and has roughly x6 less operations than MemNet (Tai et al., 2017b). It is also comparable with the CARN model in the number of operations, trading a slightly worse performance with a 2.5x size reduction. Overall, our best model is better than earlier models such as VDSR (Kim et al., 2015b) and later models such as SRMDNF (Zhang et al., 2017a) for 3x and 4x scales. Our second and third best models also outperform earlier models in performance with huge savings in the number of operations for 3x and 4x scales. Our results show that these techniques which are designed for image discriminative tasks can be effective in SR. Visual comparisons for some of these models can be found in the appendix.

## 5 QUANTIZATION AND LOW-PRECISION UNDER SUPER RESOLUTION

In our next set of experiments, we examine the viability of quantization and the use of extreme low-precision (ternary/binary) as mechanisms to reduce system resource for SR.

### 5.1 INTEGER QUANTIZATION

**Motivation:** With the success of low precision on neural networks on classification problems, we aim to show initial understanding of applying 8-bits integer quantization on our baseline model as described in section 4.1. Moving from 32-bits to 8-bits will result in a 4x reduction in memory and allow support for low-power embedded devices.

**Approach:** We train the model in full precision and apply the quantization scheme in Tensorflow-Lite for integer-only arithmetic (Jacob et al., 2017) and retrain for an additional 5 epochs with the a learning rate of $5e - 5$.

**Results:** Our results show that applying quantization lead to a slight evaluation loss in 2x scaling and a slight improvement in 4x scaling. Our results are similar to that of classification Jacob et al. (2017). Furthermore the results show that deep neural networks are robust to noise and perturbations caused by quantization. Therefore, we strongly recommend quantization especially on hardware that can further utilise its benefits.

Table 3: We extend the table that is provided by Ahn et al. (2018) and compared our best three models (in order). For fair comparisons, we do not include models that are much bigger or much smaller than our derived models.

| Scale | Model | Params (K) | Mult-Adds (G) | Set5 PSNR/SSIM | Set14 PSNR/SSIM | B100 PSNR/SSIM | Urban100 PSNR/SSIM |
|---|---|---|---|---|---|---|---|
| | VDSR (Kim et al., 2015b) | 665 | 612.6 | 37.53/0.9587 | 33.03/0.9124 | 31.90/0.8960 | 30.76/0.9140 |
| | DRCN (Kim et al., 2015a) | 1,774 | 9,788.7 | 37.63/0.9588 | 33.04/0.9118 | 31.85/0.8942 | 30.75/0.9133 |
| | LapSRN (Lai et al., 2017) | 813 | 29.9 | 37.52/0.9590 | 33.08/0.9130 | 31.80/0.8950 | 30.41/0.9100 |
| | DRRN (Tai et al., 2017a) | 297 | 6,796.9 | 37.74/0.9591 | 33.23/0.9136 | 32.05/0.8973 | 31.23/0.9188 |
| | BTSRN (Fan et al., 2017) | 410 | 207.7 | 37.75/- | 33.20/- | 32.05/- | 31.63/- |
| | MemNet (Tai et al., 2017b) | 677 | 623.9 | 37.78/0.9597 | 33.28/0.9142 | 32.08/0.8978 | 31.31/0.9195 |
| | SelNet (Choi & Kim, 2017) | 974 | 225.7 | 37.89/0.9598 | 33.61/0.9160 | 32.08/0.8984 | - |
| 2× | SRMDNF (Zhang et al., 2017a) | 2218 | 513.6 | 37.79/0.9601 | 33.32/0.9159 | 32.05/0.8985 | 31.33/0.9204 |
| | D-DBPN (Haris et al., 2018) | 1,261 | 158.9 | 38.09/0.9600 | 33.85/0.9190 | 32.27/0.9000 | 32.55/0.9324 |
| | CARN (Ahn et al., 2018) | 1,592 | 222.8 | 37.76/0.9590 | 33.52/0.9166 | 32.09/0.8978 | 31.92/0.9256 |
| | CARN-M (Ahn et al., 2018) | 412 | 91.2 | 37.53/0.9583 | 33.26/0.9141 | 31.92/0.8960 | 31.23/0.9193 |
| | **blrn-e(r=2)[rb](ours)** | **535** | **156.8** | **37.75/0.9596** | **33.30/0.9153** | **32.00/0.8973** | **31.48/0.9218** |
| | **blm1-e[rb](ours)** | **363** | **117** | **37.65/0.9592** | **33.19/0.9143** | **31.92/0.8964** | **31.13/0.9181** |
| | **blm1-e[rg](ours)** | **265** | **94.7** | **37.59/0.9590** | **33.12/0.9132** | **31.87/0.8959** | **30.91/0.9158** |
| | VDSR (Kim et al., 2015b) | 665 | 612.6 | 33.66/0.9213 | 29.77/0.8314 | 28.82/0.7976 | 27.14/0.8279 |
| | DRCN (Kim et al., 2015a) | 1,774 | 9,788.7 | 33.82/0.9226 | 29.76/0.8311 | 28.80/0.7963 | 27.15/0.8276 |
| | DRRN (Tai et al., 2017a) | 297 | 6,796.9 | 34.03/0.9244 | 29.96/0.8349 | 28.95/0.8004 | 27.53/0.8378 |
| | BTSRN (Fan et al., 2017) | 410 | 176.2 | 34.03/- | 29.90/- | 28.97/- | 27.75/- |
| | MemNet (Tai et al., 2017b) | 677 | 623.9 | 34.09/0.9248 | 30.00/0.8350 | 28.96/0.8001 | 27.56/0.8376 |
| 3× | SelNet (Choi & Kim, 2017) | 1,159 | 120.0 | 34.27/0.9257 | 30.30/0.8399 | 28.97/0.8025 | - |
| | SRMDNF (Zhang et al., 2017a) | 2,956 | 305.5 | 34.12/0.9254 | 30.04/0.8382 | 28.97/0.8225 | 27.570.8398 |
| | CARN (Ahn et al., 2018) | 1,592 | 118.8 | 34.29/0.9255 | 30.29/0.8407 | 29.06/0.8034 | 28.06/0.8493 |
| | CARN-M (Ahn et al., 2018) | 412 | 46.1 | 33.99/0.9236 | 30.08/0.8367 | 28.91/0.8000 | 27.55/0.8385 |
| | **blrn-e(r=2)[rb](ours)** | **609** | **108.1** | **34.19/0.9257** | **30.18/0.8394** | **29.00/0.8026** | **27.89/0.8469** |
| | **blm1-e[rb](ours)** | **437** | **90.5** | **34.09/0.9249** | **30.13/0.8379** | **28.95/0.8015** | **27.69/0.8419** |
| | **blrn-e(r=2)[rg+u](ours)** | **397** | **57.6** | **33.97/0.9236** | **30.04/0.8362** | **28.89/0.7997** | **27.50/0.8376** |
| | VDSR (Kim et al., 2015b) | 665 | 612.6 | 31.35/0.8838 | 28.01/0.7674 | 27.29/0.7251 | 25.18/0.7524 |
| | DRCN (Kim et al., 2015a) | 1,774 | 9,788.7 | 31.53/0.8854 | 28.02/0.7670 | 27.23/0.7233 | 25.14/0.7510 |
| | LapSRN (Lai et al., 2017) | 813 | 149.4 | 31.54/0.8850 | 28.19/0.7720 | 27.32/0.7280 | 25.21/0.7560 |
| | DRRN (Tai et al., 2017a) | 297 | 6,796.9 | 31.68/0.8888 | 28.21/0.7720 | 27.38/0.7284 | 25.44/0.7638 |
| | BTSRN (Fan et al., 2017) | 410 | 165.2 | 31.85/- | 28.20/- | 27.47/- | 25.74/- |
| | MemNet (Tai et al., 2017b) | 677 | 623.9 | 31.74/0.8893 | 28.26/0.7723 | 27.40/0.7281 | 25.50/0.7630 |
| | SelNet (Choi & Kim, 2017) | 1,417 | 83.1 | 32.00/0.8931 | 28.49/0.7783 | 27.44/0.7325 | - |
| 4× | SRDenseNet (Tong et al., 2017) | 2,015 | 389.9 | 32.02/0.8934 | 28.50/0.7782 | 27.53/0.7337 | 26.05/0.7819 |
| | SRMDNF (Zhang et al., 2017a) | 3,988 | 232.7 | 31.96/0.8925 | 28.35/0.7787 | 27.49/0.7337 | 25.68/0.7731 |
| | D-DBPN (Haris et al., 2018) | 2,207 | 79.7 | 32.47 0.8980 | 28.82/0.7860 | 27.72/0.7400 | 26.38/0.7946 |
| | CARN (Ahn et al., 2018) | 1,592 | 90.9 | 32.13/0.8937 | 28.60/0.7806 | 27.58/0.7349 | 26.07/0.7837 |
| | CARN-M (Ahn et al., 2018) | 412 | 32.5 | 31.92/0.8903 | 28.42/0.7762 | 27.44/0.7304 | 25.62/0.7694 |
| | **blrn-e(r=2)[rb](ours)** | **683** | **108.3** | **32.10/0.8938** | **28.51/0.7795** | **27.51/0.7340** | **25.95/0.7808** |
| | **blm1-e[rb](ours)** | **511** | **98.4** | **31.98/0.8921** | **28.45/0.7778** | **27.47/0.7328** | **25.80/0.7754** |
| | **blrn-e(r=2)[rg+u](ours)** | **424** | **50** | **31.84/0.8898** | **28.34/0.7750** | **27.38/0.7295** | **25.52/0.7673** |

Table 4: Quantitative results of applying 8-bit integer quantization in TF-Lite.

| Scale | Model | Params (K) | Mult-Adds (G) | Set5 PSNR/SSIM | Set14 PSNR/SSIM | B100 PSNR/SSIM | Urban100 PSNR/SSIM |
|---|---|---|---|---|---|---|---|
| 2× | bl | 961 | 231.2 | 37.82/0.9599 | 33.36/0.9160 | 32.05/0.8981 | 31.67/0.9237 |
| | bl_q | | | 37.68/0.9582 | 33.34/0.9146 | 32.01/0.8966 | 31.64/0.9226 |
| 3× | bl | 1191 | 122.4 | 34.20/0.9257 | 30.15/0.8392 | 29.01/0.8028 | 27.91/0.8467 |
| | bl_q | | | 34.17/0.9259 | 30.13/0.8393 | 29.00/0.8030 | 27.92/0.8474 |
| 4× | bl | 1154 | 93 | 32.01/0.8927 | 28.41/0.7793 | 27.49/0.7337 | 25.87/0.7792 |
| | bl_q | | | 31.99/0.8922 | 28.43/0.7794 | 27.51/0.7337 | 25.88/0.7790 |

## 5.2 TERNARY PRECISION

**Motivation:** The success of using binarized (Courbariaux & Bengio, 2016; Rastegari et al., 2016; Lin et al., 2017) and ternarized neural networks (Li & Liu, 2016; Tschannen et al., 2017) to approximate the full-precision convolutions in image discriminative tasks motivates us to experiment the effectiveness of these techniques in SR.

**Approach:** We adapt the baseline SR architecture used in prior experiments in section 4.1 but modify it structurally by replacing every convolution layer with sum-product convolution layers

proposed in StrassenNets (Tschannen et al., 2017). These sum-product convolution layers represent a sum-product network (SPN) that is used to approximate a matrix multiplication. Specifically, each convolution layer is replaced with a convolution layer that outputs $r$ feature maps, followed by a element-wise multiplication and a transpose convolution layer. As both the convolution layers hold ternary weights, the number of multiply operations required is determined by the number of element-wise multiplication which is controlled by $r$. Besides outlining the trade-off of tuning $r$, we aggressively use group convolutions.

**Results:** Similar to section 5.1, the results in Table 5 are similar to that of image discriminative tasks. Specifically, the higher the width of the hidden layer of the SPN, $r$, the better the performance at a cost of additional multiplications and additions. When $r = 6c\_out$, we achieve an evaluation score that is close to the uncompressed model for 2x scales and suffer a slight drop for 3x and 4x scales. Any further attempts to increase $r$ do not improve evaluation metric.

As proposed by Tschannen et al. (2017), we use group convolutions to reduce the number of additions. We take a step further and experiment with a wide range of groups as well. We found that the reduced number of additions do not justify the evaluation drop; the use of a lower $r$ is better than the use of groups. Additionally, since multipliers are more costly and take up more area on chip than adders, we suggest lowering $r$ instead of using grouped convolutions.

Table 5: Quantitative results of applying ternary-weighted SP convolutions. We omit the use of group convolutions as it leads to worse results. c_out refers to the number of output channels in each SP convolution layer.

| Scale | Model | r | Reduction in Mult (%) | Reduction in Add (%) | Set5 PSNR/SSIM | Set14 PSNR/SSIM | B100 PSNR/SSIM | Urban100 PSNR/SSIM |
|---|---|---|---|---|---|---|---|---|
| 2× | bl | - | - | - | 37.86/0.9600 | 33.39/0.9159 | 32.06/0.8982 | 31.74/0.9248 |
|  | ST-bl | c_out | 99.82 | -15.96 | 37.59/0.9588 | 33.14/0.9138 | 31.88/0.8959 | 31.02/0.9171 |
|  |  | 2c_out | 99.64 | -132.1 | 37.73/0.9595 | 33.30/0.9152 | 31.98/0.8973 | 31.37/0.9210 |
|  |  | 4c_out | 99.28 | -364.38 | 37.81/0.9598 | 33.31/0.9151 | 32.03/0.8978 | 31.59/0.9229 |
|  |  | 6c_out | 98.92 | -596.65 | 37.85/0.9600 | 33.41/0.9162 | 32.06/0.8981 | 31.68/0.9240 |
| 3× | bl | - | - | - | 34.24/0.9260 | 30.26/0.8405 | 29.03/0.8033 | 27.96/0.8479 |
|  | ST-bl | c_out | 99.81 | -35.58 | 33.79/0.9215 | 29.92/0.8340 | 28.82/0.7983 | 27.25/0.8318 |
|  |  | 2c_out | 99.64 | -171.33 | 33.99/0.9236 | 30.07/0.8365 | 28.91/0.8005 | 27.54/0.8389 |
|  |  | 4c_out | 99.28 | -442.83 | 34.16/0.9250 | 30.11/0.8380 | 28.97/0.8021 | 27.73/0.8435 |
|  |  | 6c_out | 98.92 | -714.33 | 34.17/0.9253 | 30.15/0.8383 | 28.99/0.8023 | 27.83/0.8454 |
| 4× | bl | - | - | - | 32.06/0.8930 | 28.49/0.7787 | 27.50/0.7337 | 25.87/0.7788 |
|  | ST-bl | c_out | 99.81 | -26.04 | 31.46/0.8829 | 28.15/0.7709 | 27.27/0.7265 | 25.24/0.7566 |
|  |  | 2c_out | 99.64 | -152.24 | 31.72/0.8871 | 28.31/0.7743 | 27.37/0.7296 | 25.49/0.7662 |
|  |  | 4c_out | 99.28 | -404.65 | 31.90/0.8901 | 28.40/0.7770 | 27.45/0.7319 | 25.67/0.7723 |
|  |  | 6c_out | 98.93 | -657.06 | 31.95/0.8907 | 28.43/0.7777 | 27.46/0.7324 | 25.77/0.7752 |

## 6  BEST PRACTICES FOR EFFICIENT SUPER-RESOLUTION

Through an extensive set of experiments, we show that some of the previous efficiency techniques that are successful in image discriminative tasks can be successfully applied to SR. Although these techniques are comparable in the former tasks, we highlight their varying effectiveness in SR and derive a list of best practices to construct or reduce any model that are designed to reduce image distortion:

- The sole use of low rank tensor decomposition (bottleneck design) results in the best trade-offs between performance and efficiency. If further compression of memory and/or compute is needed, separable/grouped convolution is recommended. If efficiency on conventional hardware is the topmost priority, we recommend reducing the number of layers or adopting the use of both channel splitting and shuffling (Ma et al., 2018).

- The fewer resource-efficient architecture changes applied, the better the trade-off. Therefore, we recommend a mixture of convolution and resource-efficient units unless further compression is needed.

- Avoid architecture changes on the first and last convolution layers.

- We strongly recommend using any form of quantization if the hardware supports it.

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

## A  VISUAL COMPARISON ON X4 SCALE BENCHMARKS

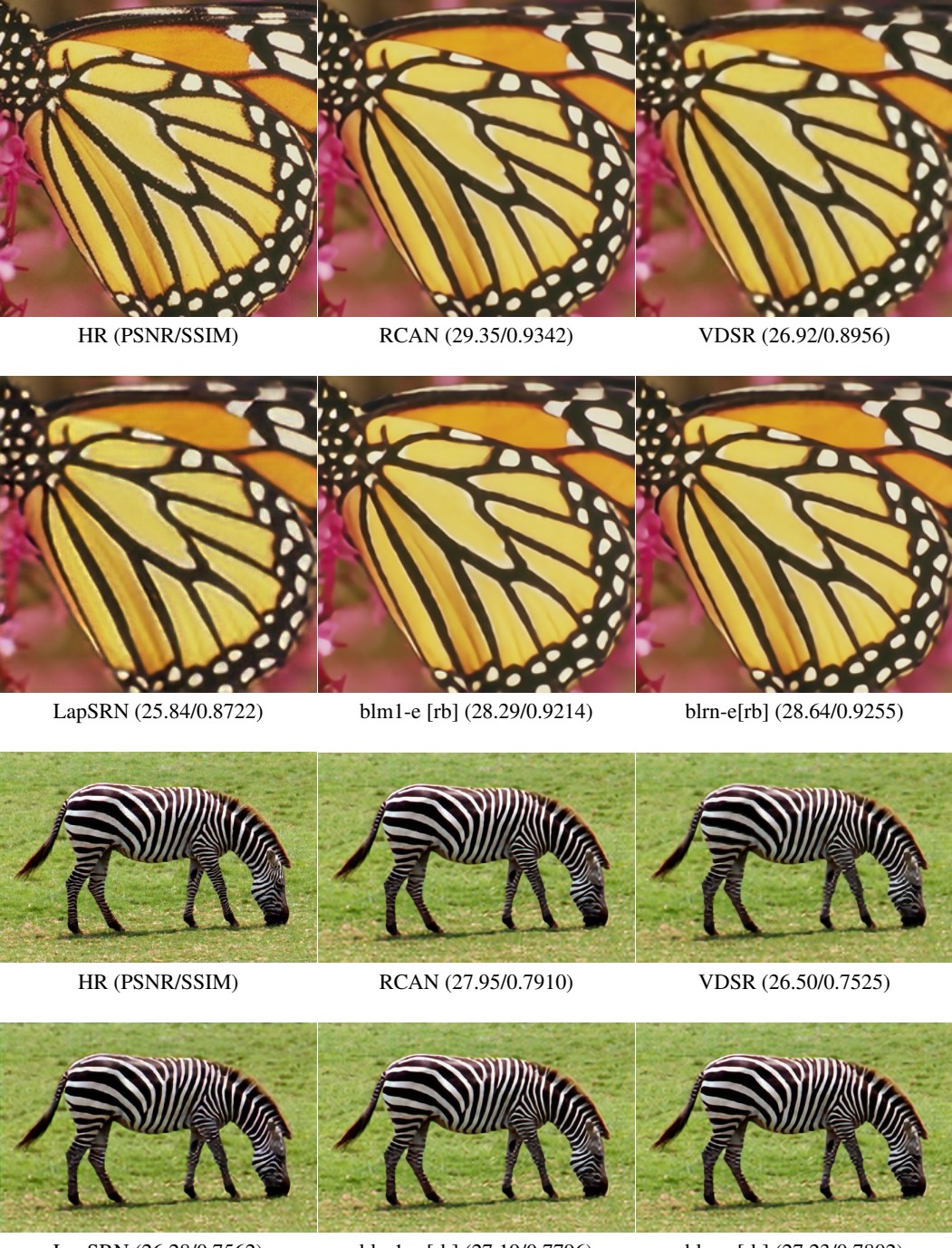

Figure 2: Visual comparisons with state-of-the-art models on *Set5* and *Set14*. VSDR and LapSRN are comparable to our models with regards to model size and/or number of operations and RCAN is x22.8-x30.5 larger and has x8.4-x9.3 more operations.

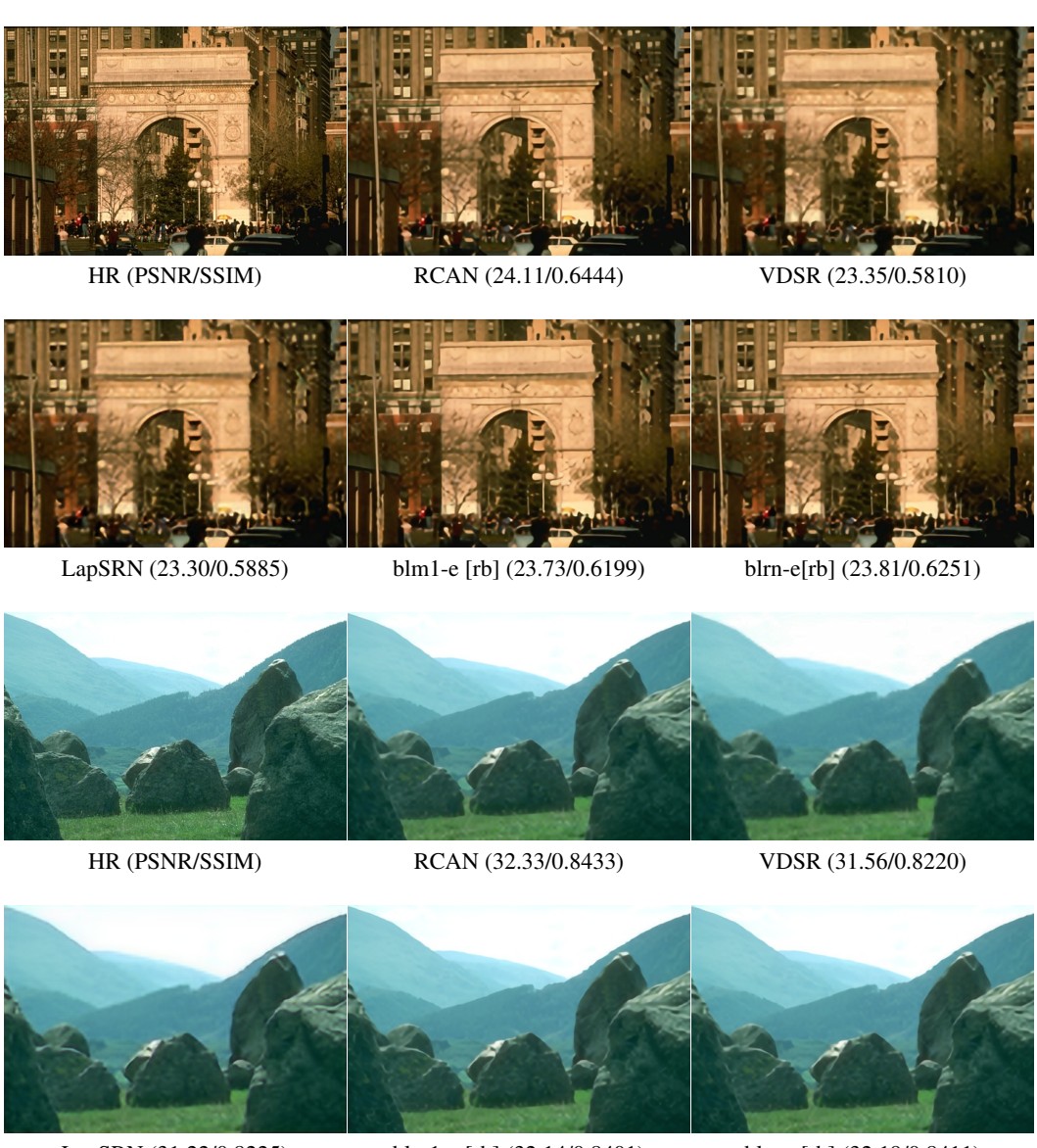

Figure 3: Visual comparisons with state-of-the-art models on *B100*.

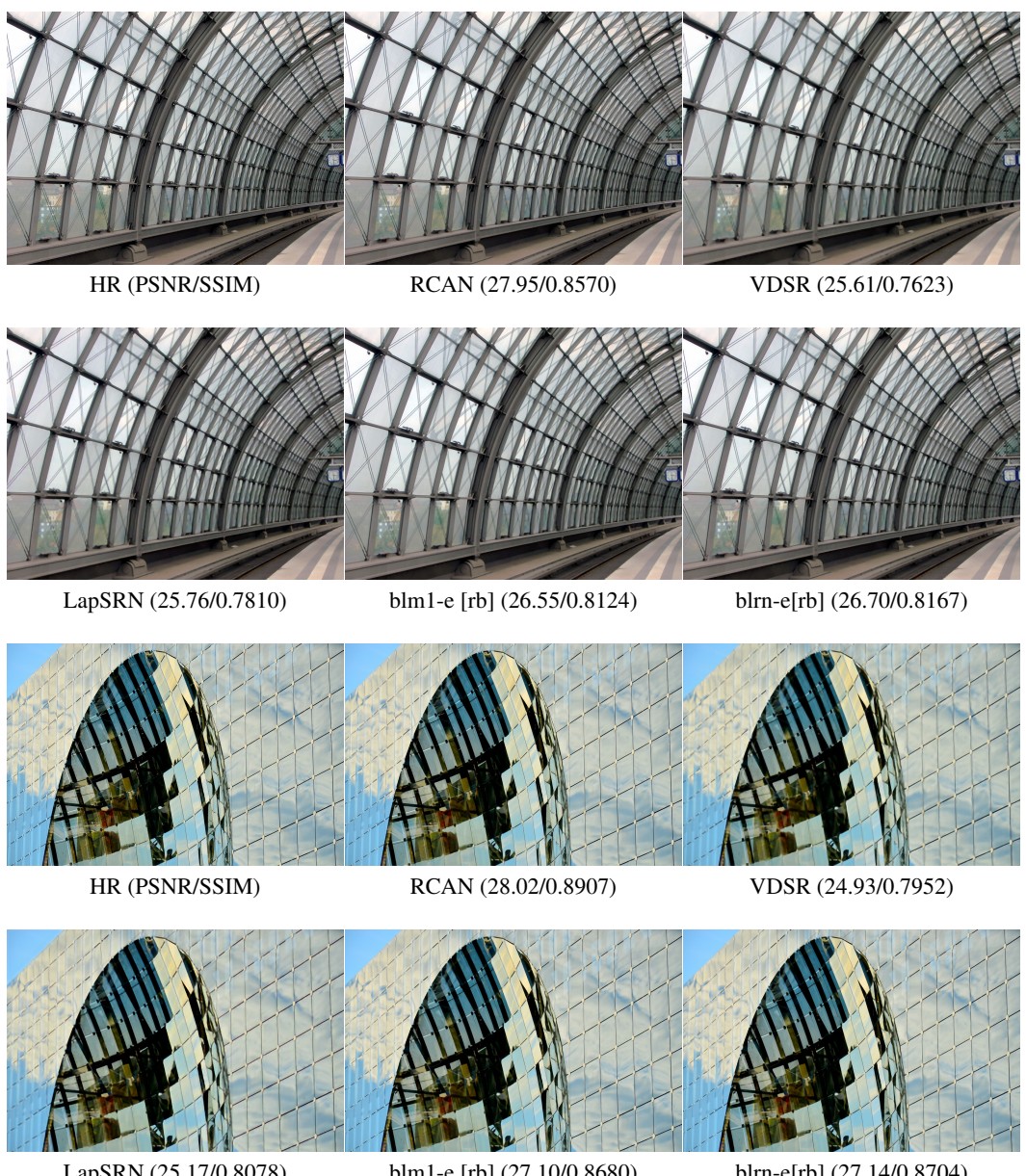

Figure 4: Visual comparisons with state-of-the-art models on *Urban100*.

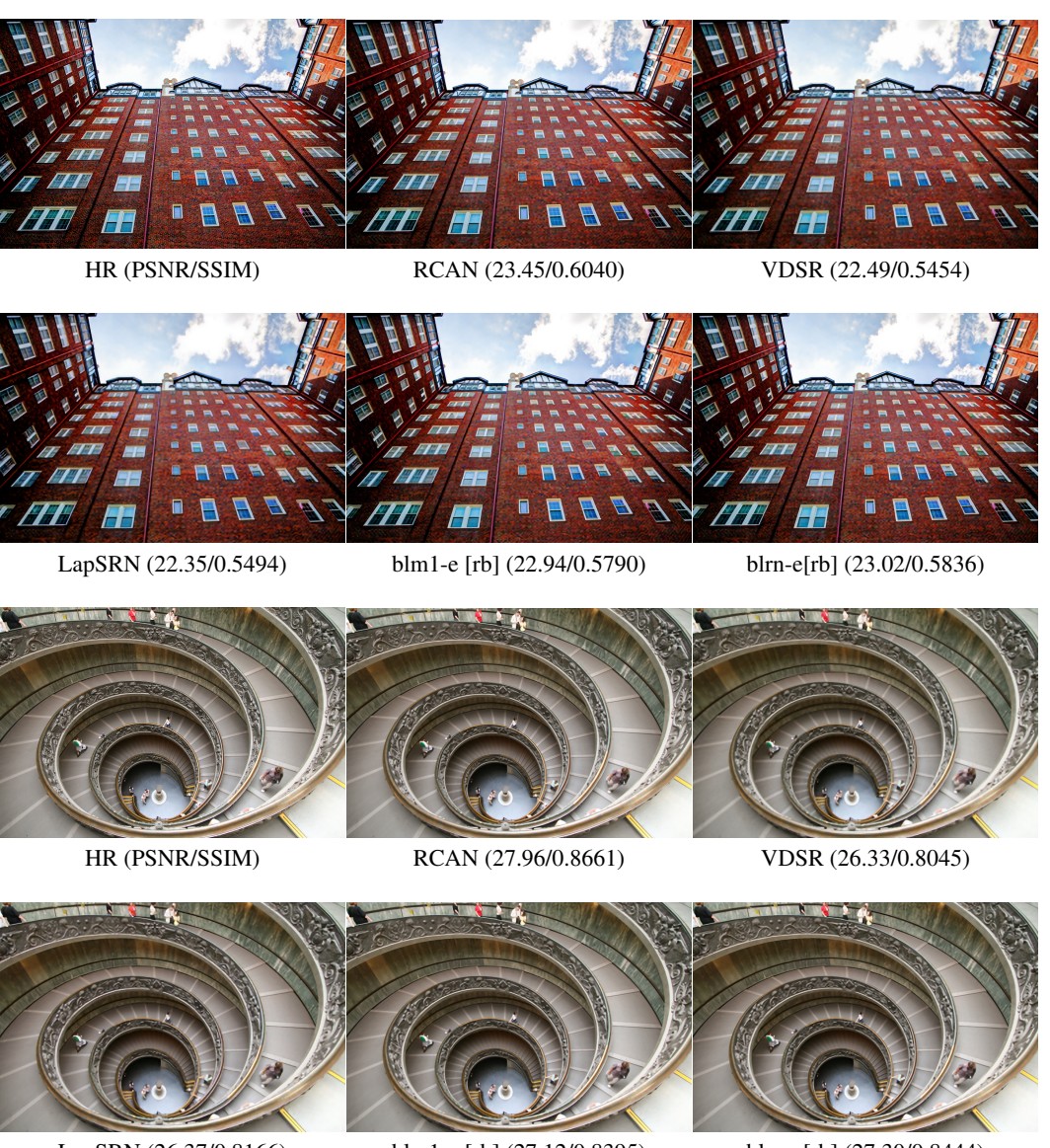

HR (PSNR/SSIM)     RCAN (23.45/0.6040)     VDSR (22.49/0.5454)

LapSRN (22.35/0.5494)     blm1-e [rb] (22.94/0.5790)     blrn-e[rb] (23.02/0.5836)

HR (PSNR/SSIM)     RCAN (27.96/0.8661)     VDSR (26.33/0.8045)

LapSRN (26.37/0.8166)     blm1-e [rb] (27.12/0.8395)     blrn-e[rb] (27.30/0.8444)

Figure 5: More x4 scale visual comparisons on *Urban100*

