# OpenReview forum: "Understanding Opportunities for Efficiency in Single-image Super Resolution Networks"
_ICLR.cc/2019/Conference_

### Official Review · AnonReviewer3 · 2018-10-30
**No new insight is proposed. The techniques are not specifically designed for super-resolution task. The experimental results are also weak.**

**Rating:** 3
**Confidence:** 5

**Review:**

This paper proposed to improve the system resource efficiency for super resolution networks.

First, I am afraid all the techniques considered in this paper have been investigated in previous works. Thus no new idea is proposed in this work. Also, it is also not clear why these improvement is particularly suitable for the task of super resolution. In my viewpoint, these techniques actually can be used to improve a variety of network architectures in both high-level and low-level vision tasks.

Second, the experimental results are also weak. As this work is aiming to address the super resolution tasks, at least visual comparisons between the proposed methods and other state-of-the-art approaches should be included in the experimental part. But unfortunately, no such qualitative results are presented in the manuscript.

Finally, the presentation of the paper should also be carefully proofread and revised.

---

> ### Author Response · Authors · 2018-11-26
> **Revised paper to underline new insights proposed and add more visual comparisons. Techniques are agnostic to any task. Better & comparable results.**
>
> Thanks for identifying the missing gaps from the paper. We have revised the paper to include more visual comparisons and make our objectives and writing clearer.
>
> The focus of the paper is to understand the empirical effects of applying and comparing existing techniques that are popular in image discriminative tasks.
>
> > All techniques considered in this paper have been investigated in previous works
>
> All techniques, apart from group convolutions which were investigated in [A], considered in this paper have not been investigated in super resolution networks. Due to the up-sampling structure of SR models, these efficiency methods may therefore produce potentially stronger side-effects to image distortion.
>
> [A] Ahn, N., Kang, B., & Sohn, K. A. (2018). Fast, Accurate, and, Lightweight Super-Resolution with Cascading Residual Network. arXiv preprint arXiv:1803.08664.
>
> > Thus no new idea is proposed in this work
>
> Although we do not propose a new way to perform compression, we show that these techniques differ greatly in trade-offs between efficiency and performance in different vision tasks, such as image classification and super-resolution. We derive a list of novel best practices from our results that can be used to efficiently construct or reduce any SR model.
>
> > Not clear why these improvement is particular suitable for the task for super resolution
>
> Low rank factorization is agnostic and is not specifically designed for any particular task. As long as the reconstruction error is small, the decomposition should follow the performance of the original model. Our results show that these techniques can also be practical and effective in super-resolution and can help existing practitioners construct or reduce their models though a list of recommendations that work better in terms of trade-off between image distortion (PSNR/SSIM) and size/operations.
>
> > these techniques actually can be used to improve a variety of network architectures in both high-level and low-level vision tasks
>
> Yes, but the extent of improvement differs in both high-level and low-level vision tasks and the trade-offs when applied to SR are unclear prior to this study. For quantization, we obtain similar trade-offs in performance and efficiency. For convolutional approximations, we show that this is not the case for different vision tasks. For instance, unlike in image classification tasks, the use of low rank tensor decomposition, which we called bottleneck reduction, has better trade-offs than the use of grouped convolutions and/or channel shuffling in super-resolution tasks. Additionally, we also show that as more layers are compressed, the worse the trade-offs, an observation which is unlike previous observations in image classification tasks.
>
> > experimental results are weak
>
> We managed to achieve better or comparable results with the models in recent existing literature. We are not aware of any model in the literature that is better in all aspects (performance, memory, and compute). To the best of our knowledge, there is always some trade-off made; if the model performs better, it is usually less efficient and vice versa. Additionally, our proposed best practices are complementary to any model in the existing literature.
>
> Once again, we would like to thank you for the time and valuable comments.

---

### Official Review · AnonReviewer1 · 2018-11-05

**Rating:** 5
**Confidence:** 4

**Review:**


The paper proposes a detailed empirical evaluation of the trade-offs achieved by various convolutional neural networks on the super resolution problem. The paper provides an extensive evaluation of different architectural changes and the trade-off between savings in terms of memory and computational cost and performance, measured in terms of PSNR and SSIM.

This is an empirical paper, thus it does not provide technical contributions. I do think that the insights obtained from such an empirical evaluation could be of interest for practitioners and researchers working on the problem. My main concern is the method only evaluates the trade-offs between model efficiency (in terms of memory and/or computation) and performance measured using metrics that are known to not be well correlated with perceptual quality. Thus it is not obvious to me that the insights obtained in this work would translate to the other case.

It is well known that PSNR favors blurry solutions over perceptually more appealing solutions. This comes from the fact that there is no information in the low resolution image to produce the missing high resolution details. Filling up plausible details in a way that is different from the original image would lead to high PSNR. Models that treat the super resolution problem as a regression task using similarity in pixel space, tend to produce blurry solutions and require very large models to improve the score.

In recent years, many works have been studying the use of perceptual losses to mitigate this issue or simply treating the super resolution problem as conditional generative modeling.  For instance, models using L2 losses in a perceptually more relevant (or learned) feature spaces [A, B], or including GAN losses [C, D] (to list a few). To my knowledge, these models are the current state of the art in terms of perceptual quality. This has been evaluated empirically via perceptual tests [D].

This line of work needs to be cited. In my view, the paper needs to provide a detailed justification on why models using these losses are not considered. Would the conclusions drawn on this work transfer to that setting? Furthermore, it would be good to perform perceptual tests to perform this evaluation. It would be good to provide some canonical examples in the appendix.

The overall writing of the paper could be improved. Several sentences are difficult to read, due to typos or the construction of the sentences. The paper evaluates many architectural modifications proposed by other works. It would be good to add an appendix with a small description of what these are. This would make the paper self-contained an easier to read (I had too look up a few of them).

The authors mentioned that they first train models for scaling factor of x2 and then use them for training settings higher magnification. How is this exactly done? Please provide details.

I am curious of weather using some for of distillation techniques would be useful here.

Did you try scaling factors larger than x4? Scaling factors of x2 does not seem very relevant, as simpler methods can achieve already quite competitive results (such as simple interpolation methods)

The authors seem to be citing Zhang et al (2018) as a reference to attention mechanisms. To my knowledge the paper that proposed these mechanisms is [E].

The citation style is not used properly throughout the manuscript. As an example:

“… proposed in StrassenNets Tschannen et al (2017).” Should be “… proposed in StrassenNets (Tschannen et al, 2017).” Or “… proposed in StrassenNets proposed by Tschannen et al (2017).”

[A] Johnson, J. et al. "Perceptual losses for real-time style transfer and super-resolution." ECCV, 2016.
[B] Bruna, J. et al "Super-resolution with deep convolutional sufficient statistics." ICLR 2016.
[C] Ledig, C. et al. "Photo-Realistic Single Image Super-Resolution Using a Generative Adversarial Network." CVPR. Vol. 2. No. 3. 2017.
[D] Sønderby, C. K., et al. "Amortised map inference for image super-resolution." arXiv preprint arXiv:1610.04490(2016).
[E] Bahdanau, D. et al "Neural machine translation by jointly learning to align and translate." arXiv (2014).

---

> ### Author Response · Authors · 2018-11-26
> **both distortion and perceptual SR have their own advantages / revised paper to show our focus in SR literature**
>
> Thank you for your valuable comments and suggestions. We revised the paper to follow your suggestions and make our points clearer.
>
> > metric that are known to not be well correlated with perceptual quality / these models are the current state-of-the-art in terms of perceptual quality
>
> We disagree. The recent PIRM 2018 Challenge [A] provided the insight that structured images look perceptually better using models that were trained to reduce image distortion (PSNR/SSIM) and unstructured details were more visually pleasing using models that were trained to improve the perception metrics which you mentioned. Therefore, we believe that both lines of work have their own advantages. Furthermore, images that are better in terms of perceptual quality performed worse than images that are better in terms of distortion quality when used as inputs for image classification [B]. Therefore, we believe that both lines of work have their own advantages.
>
> [A] Blau, Y., Mechrez, R., Timofte, R., Michaeli, T., & Zelnik-Manor, L. (2018). 2018 PIRM Challenge on Perceptual Image Super-resolution. arXiv preprint arXiv:1809.07517.
>
> [B] Jaffe, L., Sundram, S., & Martinez-Nieves, C. (2017). Super-resolution to improve classification accuracy of low-resolution images. Tech. Rep. 19, Stanford University.
>
> >this perceptual line of work needs to be cited
>
> We are aware of the perceptual track that you mentioned and only focus on the image distortion metrics. Hence, we previously kept the paper short and concise and did not cite the perceptual line of work as we did not use ideas such as the use of perpetual, contextual, adversarial losses etc. Following your advice, we have included a ‘Related work’ section to cover this and highlight the scope of our work and where it lies in the literature.
>
> > Not obvious to me that the insights obtained in this work would translate to the other case. / would the conclusions drawn on this work transfer to that setting? paper needs to provide a detailed justification on why models using these losses are not considered
>
> As mentioned, we believe that both lines of work are important. Intuitively, as the models in our experiments are not trained to improve perception metrics and the compressed super-resolution images are less visually pleasing as compared to those produced by RCAN [C], our work would not improve the score based on perceptual tests. Regardless, you made a good suggestion to use perceptual tests and we agree that it would be interesting to perform these techniques on models that are trained to improve the perception metrics or both distortion and perception metrics to look at the trade-offs. Unfortunately, doing so would involve another huge set of systematic large-scale experiments due to the large variability of how these models can be trained, a change that would be significantly different from the original scope of the paper, which focuses on the trade-offs between efficiency and the image distortion metrics.
>
> [C] Zhang, Y., Li, K., Li, K., Wang, L., Zhong, B., & Fu, Y. (2018). Image super-resolution using very deep residual channel attention networks. arXiv preprint arXiv:1807.02758.
>
> > training details
>
> We used the pretrained x2 scaling model as a starting point to train the x3 and x4 scaling models. This has been previous shown [D] to converge the model faster without affecting performance.
>
> [D] Lim, B., Son, S., Kim, H., Nah, S., & Lee, K. M. (2017, July). Enhanced deep residual networks for single image super-resolution. In The IEEE conference on computer vision and pattern recognition (CVPR) workshops (Vol. 1, No. 2, p. 4).
>
> > distillation techniques
>
> We speculate that distillation will further reduce the performance as shown in other image restoration tasks such as image enhancement [E]. However, we agree that it will be interesting as a future work to experiment and compare it with the conclusions that are proposed in our paper.
>
> [E]  Hui, Z., Wang, X., Deng, L., Gao, X.:  Perception-preserving convolutional networks for image enhancement on smartphones. In:  European Conference on Computer Vision Workshops (2018)
>
> > try scaling factors larger than x4
>
> We did not try scaling factors larger than x4. However, as the trade-offs are consistent for x2, x3, and x4 scaling factors, we strongly speculate that the same conclusions hold for high scaling factors.
>
> > simpler methods can achieve quite competitive results (such as simple interpolation methods)
>
> To the best of our knowledge, we are not aware of any simple interpolation methods that are comparable to the use of neural networks for single image super resolution.
>
> > overall writing could be improved. Citation style is not used properly.
>
> We carefully proofread and made the appropriate modifications to the paper based on your feedback. Thank you once again for the detailed review.

---

### Official Review · AnonReviewer2 · 2018-11-06
**summary of known ideas / no new ideas / no better results than other from the SR literature**

**Rating:** 4
**Confidence:** 5

**Review:**

The authors target single-image super-resolution (SR) task and study the efficiency (runtime, memory) of the current neural networks.

On the positive side, the paper is a good effort of bringing together works and insights related to efficient designs and efficient SR solutions.

If we report to the baseline architecture (RCAN) then the proposed efficient variants achieves large reductions in number of parameters or multiplications-additions, at the cost of lower accuracy. However, when the newly proposed trade-offs are compared with the existing literature, we see other methods that do a comparable or better job in the same range.

On the negative side, from my point of view, the study is far from being thorough and does not lead to or bring new insights or ideas. The experimental results does not reveal new operating points (trade-off between complexity/operations and performance accuracy).

I would suggest to the authors to expand their study, to make some novel observations, and to propose some designs that can stand out in the literature.

I am pointing out also to some recent papers that are related to the topic and can be or are applied to SR:
Gu et al, "Multi-bin Trainable Linear Unit for Fast Image Restoration Networks", arxiv 2018
Ignatov et al, "Pirm challenge on perceptual image enhancement on smartphones: Report", arxiv 2018
and some works proposed for that challenge:
Vu et al, "Fast and efficient image quality enhancement via desubpixel convolutional neural networks", ECCVW 2018
Li et al, "CARN: Convolutional Anchored Regression Network for Fast and Accurate Single Image Super-Resolution", ECCVW 2018
Pengfei et al, "Range scaling global u-net for perceptual image enhancement on mobile devices", ECCVW 2018

---

> ### Author Response · Authors · 2018-11-26
> **known ideas' outcomes is unclear on SR prior to study / new observations / better & comparable results than other**
>
> Thank you for your review and your positive comment. Our primary objective is to understand how compression techniques, that previously worked in image discriminative tasks, will work in a previously unstudied task for model compression: Super Resolution (SR). SR architectures differ significantly from those designed for image classification due to the up-sampling structure of SR models. Prior to our empirical study, it was unclear which methods that promote efficiency would perform best. Moreover, the magnitudes of gains were unknown without the extensive empirical analysis that we performed.
>
> > methods that do a comparable or better job in the same range
>
> We are not aware of any model, including those that you pointed out, that beats our best model in both efficiency (memory and compute) and the image distortion metrics (PSNR/SSIM); there is always some trade-off made. Can you give some examples on such models?
>
> > does not lead to or bring new insights or ideas. does not reveal new operating points.
>
> Our results reveal a list of new insights and operating points in terms of trade-offs between operations/size and performance accuracy that are not previously found in the SR literature:
>
> 1. In image discriminative tasks, the proposed architecture changes are comparable in terms of efficiency and accuracy trade-offs. In our work, we show varying effectiveness among these techniques. In particular, the use of low rank tensor decomposition/bottleneck reduction architectures provide the best trade-offs, followed by the use of grouped convolutions, and the use of channel shuffling & splitting. In other words, any usage of grouped convolutions increases image distortion quite significantly and any usage of channel shuffle drastically increase it even further.
>
> 2.[A, B] have shown that it is possible to maintain a similar or slight drop in performance by decomposing tensors of known models in image classification. In contrast, we show that as more tensors are decomposed in the model, the worse the trade-offs are in SR.
>
> 3. The use of ternary-weighted quantization in SR tasks results in trade-offs similar to that in image discriminative tasks. We are not aware of any other SR papers that try binary/ternary weighted architectures.
>
> [A] Bhattacharya, S., & Lane, N. D. (2016, November). Sparsification and separation of deep learning layers for constrained resource inference on wearables. In Proceedings of the 14th ACM Conference on Embedded Network Sensor Systems CD-ROM (pp. 176-189). ACM.
>
> [B] Kim, Y. D., Park, E., Yoo, S., Choi, T., Yang, L., & Shin, D. (2015). Compression of deep convolutional neural networks for fast and low power mobile applications. arXiv preprint arXiv:1511.06530.
>
> > expand their study, make some novel observations, propose some design that stand out
>
> Although we did not propose a novel design, we have made some novel observations and recommend a list of best practices for practitioners to construct or reduce any SR model in the literature.
>
> > recent papers
>
> As far as we know, the models proposed in the recent PIRM mobile challenge did not use any of the techniques that we tried and are therefore complementary to our work. Moreover, although the smaller models are more efficient, they perform much worse in terms of the image distortion metrics (PSNR/SSIM) and therefore, not a fair comparison with our derived models.
>
> We have addressed your points extensively in our revised paper attached.

---

### Meta-Review · Area_Chair1 · 2018-12-17
**lack technical contributions**

**Confidence:** 4
**Recommendation:** Reject

**Metareview:**

This paper targets improving the computation efficiency of super resolution task. Reviewers have a consensus that this paper lacks technical contribution, therefore not recommend acceptance.